# Preparation and Properties of Poly(vinyl acetate) Adhesive Modified with Vinyl Versatate

**DOI:** 10.3390/molecules28186634

**Published:** 2023-09-15

**Authors:** Guoyan Ma, Le Wang, Xiaorong Wang, Chengjun Wang, Xi Li, Lu Li, Hongfei Ma

**Affiliations:** 1College of Chemistry and Chemical Engineering, Xi’an Shiyou University, Xi’an 710065, China; 2Shaanxi Key Laboratory of Lacklustre Shale Gas Accumulation and Exploitation, Xi’an 710065, China; 3Key Laboratory of Auxiliary Chemistry and Technology for Chemical Industry, Ministry of Education, Shaanxi University of Science and Technology, Xi’an 710021, China; 4Shaanxi Collaborative Innovation Center of Industrial Auxiliary Chemistry and Technology, Shaanxi University of Science and Technology, Xi’an 710021, China; 5College of Chemistry and Chemical Engineering, Xianyang Normal University, Xianyang 712000, China; 6Department of Chemical Engineering, Norwegian University of Science and Technology, Sem Sælands vei 4, 7034 Trondheim, Norway

**Keywords:** modified poly(vinyl acetate), copolymerization, adhesives, viscosity, T-peel strength

## Abstract

A series of vinyl versatate (VV10) modified poly(vinyl acetate) adhesive (HVPVAc) were prepared using soap-free emulsion polymerization. Attenuated total reflection Fourier transform infrared (ATR-FTIR) spectroscopy was used to characterize the structure of the modified poly(vinyl acetate) latex. The effect of the VV10 content on particle size, viscosity, mechanical properties, and T-peel strength of the modified poly(vinyl acetate) was determined. No absorption peak at 1675–1500 cm^−1^ in the ATR-FTIR spectrum was observed as a result of the carbon-carbon double bond reacting completely. With the occurrence of -C-O-C and the disappearance of the carbon-carbon double bond in the FTIR spectrum, a more complex structure formed. The structure improves the mechanical properties. Increasing the VV10 content resulted in an increase in particle size from 63 nm to 221 nm, a steady increase in the viscosity of the HVPVAc latex, an increase in tensile strength from 7 MPa to 13.4 MPa, and a decrease in breaking elongation from 1310% to 1004%. As the VV10 content increased from 0 to 30% by weight, the T-peel strength of the HVPVAc adhesive increased from 8.35 N/mm to 18.97 N/mm, indicating improved adhesive performance.

## 1. Introduction

Poly(vinyl acetate) latexes are widely used in many fields, such as adhesives, coatings, paints, wood processing, construction, textiles, paper, inks, leather, and biomedical, due to their excellent waterborne properties [1,2,3] and as a result of being water-based, low cost, and the fact that they can be easily manufactured. Poly(vinyl acetate) latexes also avoid the disadvantages of organic solvent-based emulsion polymers that are toxic, flammable, and expensive. However, poly(vinyl acetate) latexes also have some drawbacks, such as poor water resistance, a higher film-forming temperature, low heat resistance, and a higher glass transition temperature. The properties of poly(vinyl acetate) latexes can be enhanced by copolymerization with other functional monomers. Vinyl chloride, vinyl propionate, vinyl versatate, butyl acrylate, 2-ethylhexyl acrylate, acrylonitrile, ethylene, and various methacrylates [4,5,6,7,8,9] have been used as functional monomers.

Vinyl versatate is rich in alkyl groups, which can introduce a significant steric hindrance. The shielding effect provided by copolymerization with vinyl versatate can improve properties such as water resistance, alkali resistance, and ultraviolet resistance [10].

Poly(vinyl acetate) latex was prepared using the traditional emulsion polymerization process. However, traditional emulsion polymerization has some disadvantages since an emulsifier is used during the reaction. To stabilize and nucleate the system, a certain amount of emulsifier was added to the reaction system, and the emulsifier was finally incorporated into the product. It is well known that the emulsifier can be difficult to remove completely, which could increase product costs. Therefore, a soap-free emulsion polymerization process may be preferable. Soap-free emulsion polymerization technology can improve the application performance of the end products as no emulsifiers are used, leading to reduced pollution of the environment. Therefore, the technique has gradually received more and more attention. Jun Wu et al. synthesized a series of poly(vinyl acetate-co-dibutyl maleate) latex particles through three methods. One of the methods was prepared in the presence of an anionic surfactant, sodium dodecyl sulfate (SDS). The other two were prepared in the presence of poly(vinyl alcohol). They compared the diffusion polymers in the prepared latex films of a latex synthesized in the presence of a surfactant with the use of poly(vinyl alcohol). The results showed remarkable differences in behavior. In the case of poly(vinyl alcohol), polymer diffusion was promoted to such an extent that the system reached its maximum extent of polymer mixing as the dispersion dried to form a film [11].

In this study, modified poly(vinyl acetate) latexes with different proportions of vinyl versatate as the modifier were prepared by soap-free emulsion polymerization. The modified structure of poly(vinyl acetate) was examined using infrared spectroscopy (IR), and the effect of the vinyl versatate content on the particle size, viscosity, mechanical properties, and T-peel strength of the modified poly(vinyl acetate) adhesive is discussed.

## 2. Results and Discussion

### 2.1. ATR-FTIR Spectroscopy of HVPVAc Films

The characteristic absorption peaks of PVA (a), PVAc (b), and HVPVAc3 (b) are shown in Figure 1. In Figure 1 a, b, and c, the characteristic peaks at 3461 cm^−1^ can be assigned to the hydroxyl groups of PVA. The stretching vibration peaks of -CH_3_ and -CH_2_- were shown at 2960 cm^−1^ and 2936 cm^−1^ [12]. The absorption peaks at 1455 cm^−1^ were due to the bending vibration of methyl. With the addition of VV10, VAc, and HEMA, some characteristic absorption peaks also appeared. Stretching vibration peaks at 1738 cm^−1^ were attributed to carbonyl groups, and the peaks at 1241 cm^−1^ belong to the -C-O-C groups [13]. The symmetric stretching vibrational absorption peaks at 1123 and 1014 cm^−1^ corresponded to the C-O bond in the ester group. As can be seen from Figure 1, the tertiary copolymer has no absorption peak at 1675–1500 cm^−1^ due to the carbon-carbon double bond, indicating that the monomers (VV10, VAc, and HEMA) react completely.

### 2.2. Nuclear Magnetic Resonance Spectrum of HVPVAc Sample

Figure 2 shows the ^1^H NMR spectrum and chemical shift of the corresponding different protons of HVPVAc3. Chemical shifts were obtained with respect to deuterated chloroform (δ = 7.28 ppm). The resonance peak at 4.00 ppm was due to the protons in -COOCH_2_. The peaks at 3.78~3.89 ppm corresponded to protons in -CH_2_CH_2_OCH_2_CH_2_-. The peaks at 2.09 ppm could be attributed to the protons in -COOCH_3_. The peaks at 1.89~1.18 ppm were assigned to the hydrogens in the main molecular chain. The singles of H in -CH_2_CCH_3_ appeared at 0.97 ppm. There were no peaks at 6.5 ppm~5.5 ppm, which indicated that the polymerization had completely occurred.

Figure 3 shows the ^13^C NMR spectrum and chemical shift of the corresponding different carbons of HVPVAc3. Chemical shifts were obtained with respect to deuterated chloroform (δ = 76.28 ppm). The peaks at 123.9 ppm were due to the carbon in -COOCH_2_. The peaks at 66.31 ppm corresponded to the carbon adjacent to the oxygen of the ester group (-COOCH_2_). The peaks at 57.45 ppm could be assigned to the carbon adjacent to the carbonyl group (CH_3_COO-). The peaks at 37.91 ppm could correspond to the carbon in -CH_2_CH_2_OCH_2_CH_2_-. The peaks at 22.39 ppm were due to the carbon adjacent to the methyl group (-CH_2_CCH_3_). The peaks at 17.96~10.57 ppm were assigned to the carbon adjacent to the main molecular chain. There were no peaks at 125 ppm~140 ppm, which also indicated that the polymerization had completely occurred. The results were consistent with the ^1^H NMR measurement.

### 2.3. Effect of the VV10 Content on the Particle Size of the HVPVAc Samples

The particle size is an important factor that influences the properties and performance of the emulsion. The size of the formed particles is mainly affected by the content of VV10 [14]. The particle stabilization mechanism is shown in Figure 4. As soon as the monomers and initiator were gradually added, polymerization was initiated. Nucleation was primarily caused by micellar nucleation. First, the free radical active chain was generated. As initiation polymerization proceeded, the basic initial particles formed through chain growth. The basic initial particles further captured the active chain of free radicals and continued to grow to form the elementary particle. The molecular chain could extend to the aqueous phase due to insufficient surface charge density. The elementary particle then coalesced, the interfacial tension decreased, and the particle stability continued to increase. Finally, stable latex particles were formed.

In this study, the effect of the VV10 content on the particle size of the HVPVAc samples is shown in Figure 5. The VV10 content had a great influence on the particle size. The z-average particle size increased from 63 nm to 221 nm as the VV10 content increased from 0 to 50% by weight.

VV10 was a very attractive monomer for the manufacture of polymers. It was a product that consisted of a mixture of vinyl esters of highly branched synthetic carboxylic acids with 10 carbon atoms. On the α-C atom, there is one methyl group and two alkyl groups [5]. It was also a hydrophobic monomer. During the copolymerization process, with the low water solubility of VV10, the nucleation was in the micellar nucleation form. With the addition of the VV10 content, the elementary nucleation particle was formed. With increasing VV10 content, the length of the molecular chain would also increase. The molecular chain then extended into the aqueous phase, and the elementary particle coalesced, which would cause the coalescent volume to increase. The higher the VV10 content, the larger the particle size.

### 2.4. Effect of the VV10 Content on the Viscosity of HVPVAc Latexes

The effect of the VV10 content on the viscosity of HVPVAc latexes is shown in Figure 6 and Table 1. Viscosity increased from 1180 mPa·s to 6580 mPa·s as the VV10 content increased from 0 to 50%. The viscosity of the HVPVAc latexes increased steadily with increased VV10 content. On the one hand, the HVPVAc latexes were prepared by terpolymerizing VAc with VV10 (an unsaturated monomer with a hydrophobic group) and HEMA. The latexes with the hydrophobic group usually have a higher viscosity [15]. VV10 was likely to laterally branch polymerization with VAc, and VV10 was rich in alkyl groups, leading to the long and complex branched chain. With the increase in VV10, the degree of entanglement of the polymer branched chains increased, thus increasing the steric hindrance of molecules and increasing the viscosity of the HVPVAc emulsion. On the other hand, the viscosity of HVPVAc latexes was also influenced by the particle size distribution. With increasing particle size, the viscosity of HVPVAc latexes also increased steadily.

### 2.5. Molecular Weight of HVPVAc Using the GPC Method

Figure 7 shows the molecular weights of HVPVAc with different VV10 contents using the GPC method. The GPC test result showed that the Mw of the series HVPVAc slowly increased from 6.19 × 10^5^ to 8.25 × 10^5^ as the VV10 content increased. The Mw results were all 10^5^ orders of magnitude. As a result of the increase in molecular weight, the molecular chain segments increase. The slower the movement of the molecular chain, the more opportunities there are for relative displacement cancellation between segments, making it difficult for the chain to release and slip. Therefore, the viscosity increased with increasing molecular weight. The molecular weight results were consistent with the viscosity results.

### 2.6. Effect of the VV10 Content on the Mechanical Properties of the HVPVAc Adhesives

Tensile strength tests were performed on different latex films containing VV10, and the effect of the VV10 content on the mechanical properties of the HVPVAc samples is shown in Figure 8. The tensile strength of the HVPVAc samples increased and the breakage elongation decreased with increasing VV10 content. With the addition of VV10 and HEMA, both reacted with VAc. With the evaporation of water, the HVPVAc films were obtained. The side chains of the HVPVAc were intertwined to produce larger and more complex structures, and the intermolecular interaction was enhanced. The molecular weights also increased, as did the forces between molecular chains. With the addition of HEMA, the terminated hydroxyls could react with each other through condensation dehydration, and then a more complex structure could be formed (the mechanism was shown in Figure 1) [16]. It may be necessary to use more force to destroy the intermolecular structure. Moreover, when films are formed through water evaporation and particle coalescence, the PVA will remain dissolved in the polymer matrix. It is presumed that this acts as a plasticizer and enhances the mechanical properties. Therefore, the tensile strength increased from 7 MPa to 13.4 MPa, and the breaking elongation decreased from 1310% to 1004%. With the VV10 content higher than 40%, the latex had an excessive viscosity, which could influence the stability of HVPVAc. Therefore, the tensile strength and the breaking elongation changed slightly.

### 2.7. Effect of the VV10 Content on the T-Peel Strength of the Samples

The results in Figure 9 show the effect of the VV10 content on the T-peel strength of the HVPVAc samples. The T-peel strength of the HVPVAc films increased from 8.35 N/mm to 18.97 N/mm when the VV10 content increased from 0% to 30 wt%. The T-peel strength of the HVPVAc samples depends on the interaction between the interior of the HVPVAc sample layer and the adhesive interface (such as van der Waals interaction, electrostatic force, and hydrogen bonding), as well as the deformation and fluidity of the adhesive. With the increase in VV10 content, the number of physical entanglement structure points between the layer and the interface increased, a dense adhesive layer was formed, and interfacial adhesion and interlayer adhesion improved. Therefore, the T-peel strength increased to a certain extent with the increase in VV10 content. With the VV10 content increasing from 30% to 50%, the T-peel strength of the HVPVAc samples decreased from 18.97 N/mm to 14.2 N/mm, which may be due to the stability of the HVPVAc latexes. The fluidity of the emulsion decreased with the excessive viscosity. The optimum VV10 content was 30%.

### 2.8. Contact Angle of HVPVAc Films with Different VV10 Content

The contact angles of HVPVAc films with different VV10 contents are described in Table 2. With the increase in the VV10 content, as shown in Table 2, the water contact angles in the HVPVAc films increased, also indicating that the HVPVAc films have some hydrophobicity. On the one hand, there are abundant alkyl groups on α-C of VV10 that form a great steric hindrance and shielding effect on the surrounding groups and protect themselves and the surrounding groups. When VV10 copolymerized with VAc and HEMA, VV10 could protect the ester groups and reduce the hydrolysis rate of the ester groups, thus improving the contact angle. On the other hand, VV10 was rich in nonpolar alkyl groups, which have strong hydrophobicity. Therefore, the contact angle of the HVPVAc films increased.

## 3. Experimental

### 3.1. Materials

Polyvinyl alcohol (PVA-1788) was supplied by Tianjin Kemiou Chemical Reagent Co., Ltd., Tianjin, China. Vinyl versatate (VV10) was produced by Shanghai Kaiyin Chemical Reagent Co., Ltd., Shanghai, China. Vinyl acetate (VAc) was obtained from Shanghai Shanpu Chemical Reagent Ltd., Shanghai, China. 2-Hydroxyethyl methacrylate (HEMA) was purchased from Tianjin Hongyan Chemical Reagent Co., Ltd., Tianjin, China. Dibenzoyl peroxide (BPO) was purchased from Tianjin Hongyan Chemical Reagent Co., Ltd., Tianjin, China.

### 3.2. Preparation of Poly(vinyl acetate) Adhesive Modified with Vinyl Versatate

PVA-1788 and deionized water were first added to a 500-milliliter, four-necked glass flask equipped with a mechanical stirrer and a thermometer. The protective colloid PVA solution was prepared at 90 °C, stirring for 2 h. The VV10, VAc, and HEMA were then added dropwise into the protective colloid solution after decreasing the system temperature to 80 °C. The BPO was also added to the system for 2 h in order to initiate the vinyl monomers. Polymerization was kept at 80 °C for 3 h. After completing copolymerization, the poly(vinyl acetate) adhesive modified with vinyl versatate (HVPVAc) latexes with different VV10 content was obtained. The bare poly(vinyl acetate) latex without the other monomers was also prepared. The typical composition was VAc/VV10 = 5:4 wt/wt (VAc + VV10 = 70 wt%), HEMA = 2 wt%, PVA = 5 wt%, and BPO = 2 wt%. The HVPVAc latexes were named HVPVAc0, HVPVAc1, HVPVAc2, HVPVAc3, HVPVAc4, and HVPVAc5 for latexes containing 0 wt%, 10 wt%, 20 wt%, 30 wt%, 40 wt%, and 50 wt% VV10 content, respectively. The HVPVAc preparation process is shown in Figure 1.

The HVPVAc films were prepared by casting the latex on a polytetrafluoroethylene disk and drying at room temperature for 3 days. Subsequently, latex films with dimensions of 200 mm × 100 mm × 1 mm were obtained. The films were stored in desiccators to avoid moisture.

### 3.3. Characterization

The chemical structure of HVPVAc samples was characterized using attenuated total reflection Fourier transform infrared (ATR-FTIR) spectroscopy. The spectra were run on a Bruker model V70 infrared spectrophotometer over 500–4000 cm^−1^ at ambient temperature with a wave number resolution of 4 cm^−1^.

NMR measurements were conducted on a BRUKER AVANCE III HD 400 M spectrometer (Germany) at 400 MHz in CDCl_3_. Tetramethylsilane (TMS) was used as an internal reference.

The particle size and particle size distribution (polydispersity index, PDI) of the HVPVAc1~5 latexes were determined by a nanoparticle size analyzer (Malvern Zetasizer Nano ZS). HVPVAc latex samples were diluted to 0.5 by weight with deionized water before measurement. Dynamic light scattering (DLS) measurements were performed before equilibrating for 120 s at 25 °C, and the mean particle size and the particle size distribution were obtained.

The viscosity measurements of the HVPVAc0~5 latexes were carried out according to GB/T 2794-1995 using an NDJ-79 Rotary Viscometer with spindle II at a rotational speed of 75 rpm at 23 ± 0.5 °C.

The gel permeation chromatography (GPC; Waters/1515, America) analysis of HVPVAc0~5 with 1% content was measured to describe the molecular weights. Tetrahydrofuran (THF) was chosen as the eluent at a flow rate of 1.00 mL/min. The weight-average molecular weight (Mw) was obtained.

The mechanical properties of the HVPVAc0~5 films were evaluated using a universal test TS2000-S machine (Yezhong Technology Co., Ltd., Yantai, China). The dimensions of the films were 200 mm × 10 mm × 4 mm. Three parallel measurements were performed to reduce the test error. The cross-head speed was 200 mm/min.

The T-peel strengths of the adhesives were studied using a universal test TS2000-S machine (Yezhong Technology Co., Ltd., Yantai, China) at ambient temperature according to GB/T 2792-2014. HVPVAc0~5 latex (0.5 g) was poured onto two rubber substrates to cover the surface uniformly and then left to dry for 1 h at 60 °C. After evaporation, a thick, solid HVPVAc adhesive film was obtained. The two rubber substrates were then placed in contact, and a pressure of 3 MPa was maintained for 3 min to ensure adhesion. The strength values of the T-peel were obtained at a 180° angle and a crosshead speed of 300 mm/min [17,18].

The static contact angles (CAs) of the distilled water HVPVAc films were measured using a JC2000A contact angle goniometer at ambient temperature using the sessile drop method. The final contact angle of each HVPVAc film was an average of at least five parallel measurements at different points.

## 4. Conclusions

The tertiary copolymer HVPVAc was successfully synthesized by the soap-free emulsion polymerization method. The particle size of the obtained copolymer can be influenced by the amount of VV10; that is, the higher the VV10, the larger the particle size. With increasing the VV10 content, the viscosity of HVPVAc latexes increased steadily, similar to the tensile strength, while the breaking elongation decreased. A pronounced phenomenon was displayed on the T-peel strength, in which it first increased and then decreased with continuously increasing the VV10 content, and the optimum content was found to be 30%. The works provided here provide one versatile method to modify poly(vinyl acetate), and it is anticipated that it can be used as an adhesive.

## Data Availability

All data generated or analyzed during this study are included in this article.

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
