# Peer review of "Preparation and Properties of Poly(vinyl acetate) Adhesive Modified with Vinyl Versatate"

_molecules, 2023, doi:10.3390/molecules28186634_

Round 1

Reviewer 1 Report

Dear Editor,

In this study, series of vinyl versatate (VV10)-modified poly(vinyl acetate) adhesive (HVPVAc) were prepared using soap-free emulsion polymerisation. The effect of VV10 content on the particle size, viscosity, mechanical properties and the T-peel strength of the modified poly(vinyl acetate) was determined. Poly(vinyl acetate) adhesive modified with vinyl versatate based polymer has different application areas. [ref. Construction and Building Materials 27 (2012) 259–262; Materials Research Vol.8, No.1, 51-56, 2005] It is available in commercial product as well. [http://www.dempolpolymer.com/dempol-va-22-nv/] The importance and novelty of this study and its contribution to the literature is limited in terms of originality. Follow of the arguments in Introduction and Result and Discussion is not well presented. A review of the English language is also highly recommended. Some other important comments given below should also be taken into consideration:

Comment 1. Abstract. “The effect of VV10 content on the particle size, viscosity, mechanical properties and the T-peel strength of the modified poly(vinyl acetate) was determined. No absorption peak at 1675- 1500 cm-1 in the ATR-FTIR spectrum was observed. “ What is the connection between the mechanical test and FTIR spectrum? Why does this region important? Abstract should be revised.

Comment 2. Part. 2.2. Preparation of poly (vinyl acetate) adhesive modified with vinyl versatate. Copolymerization studies were based on using three comonomers (The VV10, VAc and HEMA). I am wondering why HEMA (although 2%) is used as a one of the monomer in the copolymerization? It was reported that it could increase the crosslinking between the chains however to investigate the structure-property relationship, it would be useful to conduct a formulation study without HEMA. VV10 content in copolymer series were 0 wt%, 10 wt%, 20 wt%, 30 wt%, 40 wt%, and 50 wt% respectively. What is the conversion of each polymer series? Is there any residual monomer remained? Theoretical and observed feed ratio should be calculated via 1H NMR.

Comment 3. Scheme 1. Preparation process of HVPVAc. What is the abbreviation for “İa H2O”

Comment 4. Figure 4. ATR-FTIR spectrum of HVPVAc3 films. Figure 4 should be Figure 1. There is also only one film presentation. It is necessary to specify which formulation belongs to the FTIR analysis. Authors also stated that “the tertiary copolymer has no absorption peak at 1675-1500 cm-1 due to the carbon-carbon double bond, indicating that the monomers (VV10, VAc and HEMA) react completely.” However, there is slight streching in the same region in FTIR, a purification and/or an examination with 1H NMR analysis as mentioned in comment 2 is recommended. It is also recommended to measure the residual monomer amount by GC-MS analysis. This study is recommended to be done for the whole copolymer series. It would also be useful to make an overlap of one of the copolymer series with bare polyvinyl acetate in the FTIR analysis.

Comment 5. Part  3.2. Effect of VV10 content on the particle size of the HVPVAc samples. The particle stabilisation mechanism was shown in Figure 2. However, it is not clear what the colored balloons are. The relationship between the explanations general stable latex particles formation sentences at Lines 133-143 and the results obtained in the study is not clear. It was observed that the VV10 content had a great influence on the particle size. The z-average particle size increased from 63 nm to 221 nm as the VV10 content increased from 0 to 50 wt%. However, what are the reactivity ratios of each monomers. Is there any homopolymers? Could aggregation or phase separation occur when the hydrophobic monomer VV10 content increased? Large aggregates can be filtered out during the DLS analysis. It is also recommended to perform DLS analysis without using filters to compare the particles size formations. It is also recommended to purify the polymers and then applying DLS analysis.

Comment 6. Part. 3.3. Effect of VV10 content on the viscosity of HVPVAc films. No evaluation has been made about the viscosity-molecular wieght relationship. The relationship of viscosity to monomer conversion needs to be examined. It is also recommended to use/show Figure 4 or Table 1 for viscosity values.

Comment 7.  Author claim that “With the VV10 content was more than 40%, the latex had an excessive viscosity, which could influence the stability of HVPVAc” What is the relation between viscosity and stability? The effect of HEMA can be investigated in a formulation that does not contain any HEMA monomer. Related with the Comment 6, molecular weight and conversion should be investigated/compared according to the mechanical properties. Table 2 can be excluded from the manuscript.

Comment 8. Part 3.5. Effect of VV10 content on the T-peel strength of the samples. Authors claim that “The fluidity of the emulsion decreased with the excessive viscosity. The optimum VV10 content was 30%.” Is it possible to get phase separation or homopolymers of VV10 after the increasing the content in monomers feeds. Table 3 can be excluded from the manuscript.

Comment 9. Contact angle measurements can be applied for the surface analysis of the films.

A review of the English language is also highly recommended.

Author Response

In this study, series of vinyl versatate (VV10)-modified poly(vinyl acetate) adhesive (HVPVAc) were prepared using soap-free emulsion polymerisation. The effect of VV10 content on the particle size, viscosity, mechanical properties and the T-peel strength of the modified poly(vinyl acetate) was determined. Poly(vinyl acetate) adhesive modified with vinyl versatate based polymer has different application areas. [ref. Construction and Building Materials 27 (2012) 259–262; Materials Research Vol.8, No.1, 51-56, 2005] It is available in commercial product as well. [http://www.dempolpolymer.com/dempol-va-22-nv/] The importance and novelty of this study and its contribution to the literature is limited in terms of originality. Follow of the arguments in Introduction and Result and Discussion is not well presented. A review of the English language is also highly recommended. Some other important comments given below should also be taken into consideration:

Response: Firstly, we appreciate the reviewer’s kind comments. We will answer the question one by one in the following part. The corresponding changes in the manuscript were marked in red color.

  1. Abstract. “The effect of VV10 content on the particle size, viscosity, mechanical properties and the T-peel strength of the modified poly(vinyl acetate) was determined. No absorption peak at 1675- 1500 cm-1 in the ATR-FTIR spectrum was observed. “ What is the connection between the mechanical test and FTIR spectrum? Why does this region important? Abstract should be revised.

Response: We thank the reviewer’s comment. Considering the reviewer’s comment, the abstract has been revised. With the occurrence of -C-O-C and disappearance of carbon-carbon double bond in FTIR spectrum, a more complex structure formed. The structure makes the mechanical properties better. No absorption peak at 1675-1500 cm-1 in the ATR-FTIR spectrum was observed due to the carbon-carbon double bond react completely.

  1. Part. 2.2. Preparation of poly (vinyl acetate) adhesive modified with vinyl versatate. Copolymerization studies were based on using three comonomers (The VV10, VAc and HEMA). I am wondering why HEMA (although 2%) is used as a one of the monomer in the copolymerization? It was reported that it could increase the crosslinking between the chains however to investigate the structure-property relationship, it would be useful to conduct a formulation study without HEMA. VV10 content in copolymer series were 0 wt%, 10 wt%, 20 wt%, 30 wt%, 40 wt%, and 50 wt% respectively. What is the conversion of each polymer series? Is there any residual monomer remained? Theoretical and observed feed ratio should be calculated via 1H NMR.

Response: We thank the reviewer’s comment. The VV10 content is a key discussing component in investigating the structure-property relationship in this manuscript. We also prepare another manuscript in discussing the influence of HEMA content. With the addition of HEMA, the terminated hydroxyl could react with each other through condensation dehydration. Therefore, HEMA was used as a one of the monomer in the copolymerization. There was no monomer remained as the FTIR results showed. The conversion of each polymer series was nearly 100%. Considering the reviewer’s comment, the NMR test has been added in Part.3.2.

  1. Scheme 1. Preparation process of HVPVAc. What is the abbreviation for “İa H2O”

Response: We thank the reviewer’s comment. “İa H2O” means the evaporation of H2O.

  1. Figure 4. ATR-FTIR spectrum of HVPVAc3 films. Figure 4 should be Figure 1. There is also only one film presentation. It is necessary to specify which formulation belongs to the FTIR analysis. Authors also stated that “the tertiary copolymer has no absorption peak at 1675-1500 cm-1 due to the carbon-carbon double bond, indicating that the monomers (VV10, VAc and HEMA) react completely.” However, there is slight streching in the same region in FTIR, a purification and/or an examination with 1H NMR analysis as mentioned in comment 2 is recommended. It is also recommended to measure the residual monomer amount by GC-MS analysis. This study is recommended to be done for the whole copolymer series. It would also be useful to make an overlap of one of the copolymer series with bare polyvinyl acetate in the FTIR analysis.

Response: We thank the reviewer’s comment. Considering the reviewer’s comment, we have changed “Figure 4” as “Figure 1”. We’re sorry for our careless. The FTIR analysis indicates the structure of the substance. The copolymers have the similar structure, so HVPVAc3 have been used to investigate the structure. Considering the reviewer’s comment, we have marked it in the manuscript. The NMR test has been added in Part.3.2. There was no monomer remained as the FTIR results showed, so we did not use the GC-MS analysis to measure the residual monomer amount. Considering the reviewer’s comment, we will do the GC-MS analysis to measure the residual monomer amount in our future work. The bare Polyvinyl alcohol and polyvinyl acetate have been added in the FTIR analysis.

  1. Part 3.2. Effect of VV10 content on the particle size of the HVPVAc samples. The particle stabilisation mechanism was shown in Figure 2. However, it is not clear what the colored balloons are. The relationship between the explanations general stable latex particles formation sentences at Lines 133-143 and the results obtained in the study is not clear. It was observed that the VV10 content had a great influence on the particle size. The z-average particle size increased from 63 nm to 221 nm as the VV10 content increased from 0 to 50 wt%. However, what are the reactivity ratios of each monomers. Is there any homopolymers? Could aggregation or phase separation occur when the hydrophobic monomer VV10 content increased? Large aggregates can be filtered out during the DLS analysis. It is also recommended to perform DLS analysis without using filters to compare the particles size formations. It is also recommended to purify the polymers and then applying DLS analysis.

Response: We thank the reviewer’s comment. Considering the reviewer’s comment, we have added the explaination on what the colored balloons are. We have re-written the sentences at Lines 161-171. The reactivity ratios of each monomers was shown in Part. 2.2. There was no homopolymers under the reacting conditions. PVA was used the dispersant and protective colloid, so aggregation or phase separation phenomenon didn’t occur. There were no large aggregates, so filters were not used in DLS analysis. Before the DLS analysis, the purification process had been done.

  1. Part. 3.3. Effect of VV10 content on the viscosity of HVPVAc films. No evaluation has been made about the viscosity-molecular weight relationship. The relationship of viscosity to monomer conversion needs to be examined. It is also recommended to use/show Figure 4 or Table 1 for viscosity values.

Response: We thank the reviewer’s comment. Considering the reviewer’s comment, we have added the viscosity-molecular weight relationship in Line 207-215 Part.3.5. There was no monomer remained as the FTIR results showed, so we did not discuss the relationship of viscosity to monomer conversion. Considering the reviewer’s comment, we will discuss the relationship of viscosity to monomer conversion in our future work. Considering the reviewer’s comment, the viscosity values in Figure 5 and Table 1 have been added in the discussion part.

  1. Author claim that “With the VV10 content was more than 40%, the latex had an excessive viscosity, which could influence the stability of HVPVAc” What is the relation between viscosity and stability? The effect of HEMA can be investigated in a formulation that does not contain any HEMA monomer. Related with the Comment 6, molecular weight and conversion should be investigated/compared according to the mechanical properties. Table 2 can be excluded from the manuscript.

Response: We thank the reviewer’s comment. When the viscosity of HVPVAc was higher, the flowability of the latex became poor, and the latex may be formed gels, which could be an factor influencing the latex stability. Considering the reviewer’s comment, another manuscript in discussing the influence of HEMA content will be prepared. The relationship of molecular weight to the mechanical properties was added in Line 225-226 Part.3.6. Considering the reviewer’s comment, Table 2 was excluded from the manuscript.

  1. Part 3.5. Effect of VV10 content on the T-peel strength of the samples. Authors claim that “The fluidity of the emulsion decreased with the excessive viscosity. The optimum VV10 content was 30%.” Is it possible to get phase separation or homopolymers of VV10 after the increasing the content in monomers feeds. Table 3 can be excluded from the manuscript.

Response: We thank the reviewer’s comment. The crosslinking structure may be increased after increasing the content in monomers feeds, and which may be the major reason in the excessive viscosity. Considering the reviewer’s comment, Table 3 was excluded from the manuscript.

  1. Contact angle measurements can be applied for the surface analysis of the films.

Response: We thank the reviewer’s comment. Considering the reviewer’s comment, we have added the contact angle measurements in Part.3.7.

In this updated version, we have gone through the manuscript and modified the content.

At last, we thank the reviewer and editor again for the kind comments.

Reviewer 2 Report

Dear Authors,

Line 124: In the formulation, there is HEMA present, whose -OH group can contribute to this absorption band

Line 100 to110: please indicate the acronyms used in Figure 2 for the measured quantities.

Figure 3: There is something wrong with the graphs. For example, the peak of HVPVAc3 is to the left of that of HVPVAc1 indicating a lower particles size, but the reported data of Dz do not match.

Figure 4: please remove the word "films." The measurement was performed on emulsions, not on films.

It's not clear the PVA contribution. Since it doesn't participate in the reaction, when films are formed through water evaporation and particle coalescence, the PVA will remain dissolved in the polymer matrix. It is presumed that this acts as a plasticizer, affecting the mechanical properties. At the same time, not being bound to the polymer matrix, it can be expected to migrate outward over time. Alternatively, it could be extracted if the films come into contact with water. I suggest that the authors duly consider these aspects.

Author Response

Firstly, we appreciate the reviewer’s kind comments. We will answer the question one by one in the following part. The corresponding changes in the manuscript were marked in red color.

  1. Line 124: In the formulation, there is HEMA present, whose -OH group can contribute to this absorption band

Response: We thank the reviewer’s comment. The -OH group can be contributed to PVA. With the addition of HEMA, the terminated hydroxy could react with each other through condensation dehydration, then a more complex structure maybe formed as shown in Scheme 1.

  1. Line 100 to110: please indicate the acronyms used in Figure 2 for the measured quantities.

Response: We thank the reviewer’s comment. Considering the reviewer’s comment, we have added the acronyms in Line 107 to 135 and Figure 3.

  1. Figure 3: There is something wrong with the graphs. For example, the peak of HVPVAc3 is to the left of that of HVPVAc1 indicating a lower particles size, but the reported data of Dz do not match.

Response: We thank the reviewer’s comment. We are sorry for misplacing the graphs. We have checked and redraw all the graphs.

  1. Figure 4: please remove the word "films." The measurement was performed on emulsions, not on films.

Response: We thank the reviewer’s comment. Considering the reviewer’s comment, the word "films" was removed.

  1. It's not clear the PVA contribution. Since it doesn't participate in the reaction, when films are formed through water evaporation and particle coalescence, the PVA will remain dissolved in the polymer matrix. It is presumed that this acts as a plasticizer, affecting the mechanical properties. At the same time, not being bound to the polymer matrix, it can be expected to migrate outward over time. Alternatively, it could be extracted if the films come into contact with water. I suggest that the authors duly consider these aspects.

Response: We thank the reviewer’s comment. Considering the reviewer’s comment, we have added the reviewer’s comment in Line 231-233 Part.3.6.

At last, we thank the reviewer and editor again for the kind comments.

Round 2

Reviewer 1 Report

Author replied to all comments and gave brief information. However, NMR analysis should be analyzed once more. Figure 2 showed the chemical shift of 13C NMR and the assignment of corresponding different carbons of HVPVAc3. It is seen that monomer remains in this spectrum. Monomer signals look sharp. The carbonyl ester of the polymer is not visible in the spectrum as well. It is also recommended to give 1H NMR analysis with 13 CNMR. It is not enough to state that there is no monomer left according to the FTIR analysis. It is also recommended to review the discussion after the NMR results.

Professional editing is suggested. 

Author Response

Author replied to all comments and gave brief information. However, NMR analysis should be analyzed once more. Figure 2 showed the chemical shift of 13C NMR and the assignment of corresponding different carbons of HVPVAc3. It is seen that monomer remains in this spectrum. Monomer signals look sharp. The carbonyl ester of the polymer is not visible in the spectrum as well. It is also recommended to give 1H NMR analysis with 13 CNMR. It is not enough to state that there is no monomer left according to the FTIR analysis. It is also recommended to review the discussion after the NMR results.

Response: Firstly, we appreciate the reviewer’s kind comments. The corresponding changes in the manuscript were marked in red color. Considering the reviewer’s comment, we have done the NMR test again. The 1H NMR and 13CNMR analysis were both shown in Line 151-170 Part.3.2. The discussion after the NMR results were also modified.

At last, we thank the reviewer and editor again for the kind comments.